# Construction of Z-Scheme TiO_2_/Au/BDD Electrodes for an Enhanced Electrocatalytic Performance

**DOI:** 10.3390/ma16020868

**Published:** 2023-01-16

**Authors:** Kai Zhang, Kehao Zhang, Yuxiang Ma, Hailong Wang, Junyong Shao, Mingliang Li, Gang Shao, Bingbing Fan, Hongxia Lu, Hongliang Xu, Rui Zhang, Huanhuan Shi

**Affiliations:** 1School of Materials Science and Engineering, Zhengzhou University, Zhengzhou 450001, China; 2Zhongyuan Critical Metals Laboratory, Zhengzhou 450001, China; 3State Key Laboratory of Superabrasives, Zhengzhou Research Institute for Abrasives & Grinding Co., Ltd., Zhengzhou 450001, China; 4School of Material Science and Engineering, Luoyang Institute of Science and Technology, Luoyang 471023, China; 5School of Ecology and Environment, Zhengzhou University, Zhengzhou 450001, China

**Keywords:** boron-doped diamond, TiO_2_, Z-scheme, electrocatalysis, wastewater treatment

## Abstract

TiO_2_/Au/BDD composites with a Z-scheme structure was prepared by orderly depositing gold (Au) and titanium dioxide (TiO_2_) on the surface of a boron-doped diamond (BDD) film using sputtering and electrophoretic deposition methods. It was found that the introduction of Au between TiO_2_ and the BDD, not only could reduce their contact resistance, to increase the carrier transport efficiency, but also could improve the surface Hall mobility of the BDD electrode. Meanwhile, the designed Z-scheme structure provided a fast channel for the electrons and holes combination, to promote the effective separation of the electrons and holes produced in TiO_2_ and the BDD under photoirradiation. The electrochemical characterization elucidated that these modifications of the structure obviously enhanced the electrocatalytic performance of the electrode, which was further verified by the simulated wastewater degradation experiments with reactive brilliant red X-3B. In addition, it was also found that the photoirradiation effectively enhanced the pollution degradation efficiency of the modified electrode, especially for the TiO_2_/Au/BDD-30 electrode.

## 1. Introduction

Electrocatalytic oxidation technology [1,2,3] is exhibiting increasing application prospects in the field of wastewater treatment [4,5,6,7,8], by virtue of its advantages of the high efficiency mineralization of organic pollutants, flexible operation, and no addition of chemicals [9,10,11]. Electrode materials are the key components that determine its environmental application, and a boron-doped diamond (BDD) has been reported as one of the most promising anodes to oxidize the recalcitrant and complex wastewater [12,13,14,15]. However, the high production cost and relatively low electrocatalytic efficiency hinder its large-scale application. To improve the catalytic efficiency and reduce the production cost of the BDD, many efforts have been made, and it was found that the construction of composite electrodes, via the introduction of other semiconductor materials, was an efficient breakthrough [16,17,18]. Titanium dioxide (TiO_2_), as an ultraviolet response material, has been widely studied in the photocatalysis field [19,20,21]. Thus, the combination of TiO_2_ and the BDD has been explored by researchers, to obtain a novel composite electrode [22,23,24]. Fernando et al. [25] found that the TiO_2_/BDD composite electrodes prepared by the electrophoretic deposition could significantly enhance the degradation rate of Acid Blue 80 under the photoelectric synergy. Yu et al. [26]. reported that the TiO_2_/BDD composite electrodes fabricated by the metal-organic chemical vapor deposition system could efficiently degrade the reactive yellow 15 and reduce the hexavalent chromium under the UV light irradiation. Wei et al. [27] found that a Au/TiO_2_ nanorod modified the BDD electrode obtained by the chemical reduction method showed the favorable electro-catalytic activity toward the detection of catechol with a fast response, a high sensitivity, and a low detection limit, as compared with the bare BDD electrode. Therefore, the TiO_2_ modification of the electrode is a potentially effective way to enhance the electrocatalytic performance of the BDD. However, the poor adhesion of TiO_2_ on the surface of the BDD still need to be optimized by the exploration of new methods, to further improve the catalytic performance and the practical application of the BDD electrodes.

Studies have proven that the Z-scheme structure could promote the photocatalytic ability of the photocatalyst, by inhibiting the recombination of the photoinduced electrons and holes [28,29,30]. As reported, the all-solid Z-scheme TiO_2_-Au-CdS system [31] had a much better catalytic activity than that of the relative two-components Au-TiO_2_ and TiO_2_-CdS specimens. The AgI/Ag/AgBr composites [32] with the Z-scheme structure, prepared using a facile in-situ ion exchange method along with light reduction, displayed an excellent photocatalytic activity in the degradation of methyl orange under visible light irradiation (λ > 420 nm). In addition, Wang et al. [33] found that the metal Cd core in the ZnO-Cd-CdS catalyst acted as an efficient charge-carrier transport channel under solar light illumination, to boost the photocatalytic hydrogen (H_2_) evolution rate. However, the function mechanisms of the Z-scheme structure in electrocatalysis or the combination of electrocatalysis and photocatalysis are still unknown, especially the BDD based Z-scheme system for electrocatalysis, is more rarely reported.

Herein, the Z-scheme TiO_2_/Au/BDD composite electrodes were designed and prepared to enhance the application potential of the BDD in the field of wastewater treatment. In this study, the electrochemical properties of the TiO_2_/Au/BDD composite electrodes with a different Au content were detected. The photo-electrocatalytic performance of the TiO_2_/Au/BDD composite electrodes was studied, based on the degradation experiments of the reactive brilliant red X-3B. This work indicated that the Au layer could improve the photo-electrocatalytic performance of the TiO_2_/Au/BDD composite electrodes.

## 2. Materials and Methods

### 2.1. Preparation of the TiO_2_/Au/BDD Electrode

The silicon-based BDD electrode (Zhengzhou Research Institute for Abrasives & Grinding Co., Ltd., China) was cleaned ultrasonically with acetone, absolute ethanol, and deionized water, in turn. Then, Au was deposited on the surface of the oven-dried BDD via an ion sputtering apparatus (SBC-12, KYKY Technology Co., Ltd., Beijing, China), to obtain the Au/BDD electrode. The sputtering current was 4 mA. The deposition amount of Au was controlled by the different sputtering time. The sputtering time of Au was 30 s, 60 s, and 90 s, separately. 

To prepare the TiO_2_ sol, 15 mL of Tetrabutyl titanate (TBOT, Shanghai Aladdin Bio-Chem Technology Co., Ltd., China) and 33 mL of absolute alcohol were mixed in the first beaker and stirred for 30 min. Subsequently, 4.5 mL of deionized water and 15 mL of absolute ethanol were mixed in the second beaker and stirred for 10 min. The mixture of deionized water and absolute ethanol was added into the first beaker drop by drop, and the pH value of the mixture was adjusted to 3 by glacial acetic acid. 

Furthermore, TiO_2_/Au/BDD electrode was fabricated by the electrophoretic deposition method. The electrophoretic deposition was carried out in TiO_2_ sol with the Au/BDD cathode and graphite anode. A 40 V bias voltage was applied using a DC power source (MS1001D, Dongguan Maihao Electronic Technology Co., Ltd., China). Following the deposition, for 60 s, the electrode was immediately heated at 450 °C for 1 h, to obtain TiO_2_. The deposition process was repeated three times. According to the different sputtering time of Au in the Au/BDD electrode, the obtained electrodes were labelled as TiO_2_/Au/BDD-30, TiO_2_/Au/BDD-60, and TiO_2_/Au/BDD-90, respectively. For comparison, the TiO_2_/BDD electrode was also prepared by the above method without the deposition of Au and using the bare BDD as the cathode in the electrophoretic deposition.

### 2.2. Simulated Wastewater Degradation Experiment

The photo-electrocatalytic degradation experiments were conducted in a beaker, as described in the schematic illustration in Figure 1. The anode (BDD, TiO_2_/BDD, and TiO_2_/Au/BDD-30) and the graphite cathode were placed in parallel, and the photoirradiation was launched from the top side using a Xe lamp source (PLS-SXE300, Beijing perfectlight Technology Co., Ltd., China) to serve as the simulated solar light. For the experiment, a 200 mL of 100 mM Na_2_SO_4_ aqueous solution containing 200 mg/L reactive brilliant red X-3B (Shanghai Aladdin Bio-Chem Technology Co., Ltd., China) was continuously stirred in the reactor, and a constant current of 0.6 A was applied to the electrodes. The samples were collected at the preselected time intervals to quantify the concentration of the reactive brilliant red X-3B at 538 nm by UV-visible absorption photometer (UV-1800PC, Shanghai Mapada Instruments Co., Ltd., China). The decolorization rate was further calculated according to the absorbance values, using the following equation [34]:(1)$$\eta \left(\%\right)=\frac{\left(A_{0}-A_{1}\right)}{A_{0}}\times 100\%$$
where η is the decolorization rate, A_0_ and A_1_ are the absorbance values of the wastewater before and after the treatment for a certain period.

### 2.3. Characterization

The phase composition of the electrodes was performed via X-ray diffractometer (XRD, PANalytical EMPYREAN, Malvern Panalytical Ltd., Malvern, UK). The structure information of the electrodes was carried out by Laser Raman spectrometer (LabRAM HR Evolution, HORIBA France SAS, Paris, France) with a 532 nm laser. The morphology and elemental distribution of the electrodes were analyzed by a field-emission scanning microscope (FESEM, JSM-6700F, JEOL, Tokyo, Japan) with an energy dispersive spectrometer (EDS). The surface resistivity, carrier concentration, and the Hall mobility of the electrodes were determined via the Hall measurement system (LakeShore 8400, Lake Shore Cryotronics, Inc., Westerville, OH, USA). The electrochemical properties of the electrodes were obtained by an electrochemical analyzer (CHI604E, Shanghai Chen Hua Technology Co., Ltd., China).

## 3. Results

### 3.1. Phase Analysis of the Electrode

To examine the crystallographic structure of the obtained electrodes, the XRD patterns were carried out. As shown in Figure 2, two obvious peaks at 43.97° and 75.26° were observed in the XRD spectrum of the bare BDD, corresponding to the (111) and (220) crystal planes of the diamond (JCPDS 01-089-3441). The other two characteristic peaks at 28.60° and 47.34° were attributed to the Si substrate (JCPDS01-075-0590). For the TiO_2_/Au/BDD composite electrodes, the characteristic peaks of the (111), (220), and (311) crystal planes of Au (JCPDS01-089-3697) were detected at 38.17°, 64.59°, and 77.69°, in the spectrum, and the peak at 25.43° corresponds to the (101) crystal plane of anatase TiO_2_ (JCPDS 00-004-0477). These XRD results indicated that both Au and TiO_2_ were successfully deposited on the BDD electrode and the characteristic peaks intensities of Au in the TiO_2_/Au/BDD electrodes gradually increased with the increase of its loading content.

The Raman spectroscopic investigation was also conducted to explore whether the BDD was graphitized during the heat treatment, and the results are presented in Figure 3. The sharp peak at 1331.65 cm^−1^ agreed with the standard Raman peak of the diamond (1332 cm^−1^). The reason for the slight downshift of the Raman band may be the doping of boron. Meanwhile, no characteristic peak of the graphite structure was found [35], indicating that the significant graphitization of the electrodes did not occur in the heat treatments. In addition, the fluorescence interference in the Raman spectra of the composite electrodes was observed, and the interference intensity enhanced with the increasing of the deposition amount of Au [36].

### 3.2. Morphology and the Element Distribution Analysis of the Electrode

The morphology of the bare BDD and the TiO_2_/Au/BDD electrodes are shown in Figure 4. Except for the bare BDD shown in Figure 4a, the other electrodes containing TiO_2_ had a good glossiness. This indicated that the electrophoretic deposition method was an effective way to improve the surface morphology of the electrode. From these images shown in Figure 4c–e, there was no obvious difference among the TiO_2_/Au/BDD electrodes. Moreover, the heat treatment of 450 °C for 1 h was acceptable for the electrodes because no cracks were found on the surface.

In order to determine the existence of TiO_2_ and Au on the surface of the BDD electrode, the EDS element mapping images were performed, as shown in Figure 5b–e and the atomic percentages are shown in Figure 5f. According to the images, Au, Ti, and O elements were distributed uniformly without obvious aggregation on the surface, indicating that TiO_2_ and Au were successfully deposited on the surface of the bare BDD. Therefore, the sandwich-type TiO_2_/Au/BDD composite structure was constructed as expected in this work.

### 3.3. Electrochemical Performance Analysis

The Hall tests of these various electrodes were carried out and the results are summarized in Table 1. As seen, the Hall mobility, the sheet carrier concentration, and the sheet resistivity of the bare BDD were 158 cm^2^/V·s, 2.26 × 1 017 cm^−2^, and 0.175 Ωm, respectively. In addition, the three performance parameters of the electrodes changed obviously, after the deposition of TiO_2_ and Au. The Hall mobility, the sheet carrier concentration, and the sheet resistivity of the TiO_2_/BDD were 178 cm^2^/V·s, 2.03 × 1017 cm^−2^, and 0.173 Ωm, respectively. For the TiO_2_/Au/BDD electrodes, the Hall mobility of the electrodes increased, the sheet carrier concentration and the sheet resistivity decreased with the increase of the deposition amount of Au. The Hall mobility, the sheet carrier concentration, and the sheet resistivity of the TiO_2_/Au/BDD-30 were 194 cm^2^/V·s, 2.25 × 1017 cm^−2^, and 0.143 Ωm, respectively.

Furthermore, the electrochemical properties of these electrodes were measured by cyclic voltammetry (CV) tests in acidic, alkaline, and neutral conditions, respectively. Figure 6 illustrated that the electrochemical potential windows of the bare BDD electrode in 0.1 mol/L H_2_SO_4_, 0.1 mol/L NaOH, and 0.1 mol/L Na_2_SO_4_ solution were 2.52 V, 2.63 V, and 3.14 V (vs. standard hydrogen electrode, SHE), respectively. For the TiO_2_/BDD electrode, the potential windows showed a slight enhancement in all of the three electrolytes, and the values were 2.65 V, 2.89 V, and 3.19 V (vs. SHE), respectively. For the TiO_2_/Au/BDD electrodes, all of the potential windows in the different electrolytes were lower than those of the BDD and TiO_2_/BDD electrodes, and they shifted down continuously in all electrolytes as the deposition amount of Au increased. The results suggested that the introduction of TiO_2_ could broaden the potential window of the BDD electrode. On the contrary, Au narrowed the potential window values of the TiO_2_/BDD electrode. Moreover, it was also found that the potential window values in the neutral environment were wider than those in both the acidic and alkaline conditions. This may result from the acidic and alkaline solution that contained a large amount of H^+^ and OH^−^, which was conducive to the production of hydrogen and oxygen, so that the oxygen evolution and hydrogen evolution reaction occurred at a lower potential [37]. The electrochemical properties of the electrodes, compared with other studies, are shown in Table 2. As shown, the oxygen evolution potential and potential window of the electrodes, in the literature, were all tested in an acidic medium, thus the equivalent comparison was difficult to achieve. In this study, although Au narrowed the oxygen evolution potential and the potential window values of the electrodes, the photo assisted electrochemical treatment efficiency of the TiO_2_/BDD electrode was enhanced by the Au layer, which helps to promote the practical application potential of the BDD in the field of wastewater treatment.

The CV curves in a mixture solution of 5 mM K_3_Fe(CN)_6_ and 1 M KCl, were further measured at the scan rate of 50 mV/s, and are shown in Figure 7. According to the results, all electrodes had lower redox properties. It was worth noting that the optimal sputtering time of Au was 30 s because the TiO_2_/Au/BDD-30 exhibited the largest redox peak area in all electrodes. However, as the deposition amount of Au increased, the redox properties of TiO_2_/Au/BDD-60 and TiO_2_/Au/BDD-90 electrodes decreased and even became weaker than that of the bare BDD electrode. This phenomenon could be explained by the rough morphology of the Au film and the larger Au particles, which caused a poor contact between TiO_2_ and the BDD in the process of sputtering and the heat treatment that lead to the appearance of the lower conductivity of the electrodes [40].

### 3.4. Treatment of the Simulated Wastewater

The simulated wastewater with 200 mg/L reactive brilliant red X-3B was treated using the prepared electrodes as anode, in a batch reactor under a constant current of 0.6 A, and the effect of the photoirradiation on the pollutant degradation was also measured. As shown in Figure 8, the decolorization rate was calculated by Equation (1). The continuous degradation of the reactive brilliant red X-3B by the bare BDD was evident with the extension of the electric current time applied, and the near complete removal was achieved at 1 h. Interestingly, both TiO_2_/BDD and TiO_2_/Au/BDD-30 electrodes exhibited a significantly better electrooxidation efficiency than the bare BDD, which was consistent with the analysis results of the electrochemical performance above. In addition, it was also found that the degradation rates of the reactive brilliant red X-3B by TiO_2_/BDD and TiO_2_/Au/BDD-30 electrodes, were effectively improved with the participation of the photoirradiation, as described in the experimental section. For example, the treatment of the simulated wastewater for 30 min in the photo-electrocatalytic experiment, the removal rates of reactive brilliant red X-3B by the three electrodes, were significantly different under the same conditions. The removal rates of the bare BDD electrode with and without the photoirradiation were only 75% and 74%, respectively. Following the introduction of TiO_2_, the removal rates of the TiO_2_/BDD electrode with and without photoirradiation were 92% and 89%, respectively. It was remarkable that the removal rates of the TiO_2_/Au/BDD-30 electrode with and without photoirradiation were 98% and 93%, respectively.

### 3.5. The Potential Working Mechanism of the TiO_2_/Au/BDD Electrode

According to the experimental results above and the previous studies, the working mechanism of the TiO_2_/Au/BDD electrode was proposed. Figure 9 shows the schematic diagram of the TiO_2_/Au/BDD Z-scheme electron transfer process, where Au acts as the electron mediator, namely, the insertion of Au between TiO_2_ and the BDD forms the known Ohmic contact with a low contact resistance [41,42,43]. That means that the electrons produced in the conduction band (CB) of TiO_2_ can directly combine with the holes produced in valence band (VB) of the BDD through the Ohmic contact, which helps the separation and transportation of e^−^ and h^+^ pairs in TiO_2_ and the BDD. Therefore, the residue VB-holes of TiO_2_ and CB-electrons of the BDD can directly or indirectly react with the pollutants in the wastewater, such as, h+, not only can oxidize H_2_O or OH^-^ into hydroxyl radical (•OH) to the degradation pollutants, but also directly react with the pollutants, and the photoinduced e^−^ can react with O_2_ to generate O^2^.^-^ to remove pollutants [44]. Therefore, combined with the electrochemical performance analysis, the TiO_2_/Au/BDD-30 electrode is a highly qualified candidate in the application of the photo assisted electrochemical treatment system, which was also proved in the degradation of the reactive brilliant red X-3B.

## 4. Conclusions

In this study, the TiO_2_/Au/BDD electrodes with a different deposition amount of Au, were fabricated and used to explore their photo-electrochemical performance in the field of pollutant degradation. The results demonstrated that the sandwich-type Z-scheme TiO_2_/Au/BDD composite structure was successfully constructed, where a significant graphitization in the heat treatment did not occur. Benefiting from the unique structure and the charge transport channel of Au, the TiO_2_/Au/BDD electrode exhibited a much greater electrocatalytic oxidation and photodegradation performances than the bare BDD and TiO_2_/BDD composite electrodes. Au provided to be a fast channel for the carrier transport, so that the electrons (e^-^) of TiO_2_ and holes (h^+^) of the BDD recombined in the Au layer, effectively separating the e^-^ and h^+^ inside TiO_2_ and the BDD, thereby enhancing the electrocatalytic redox performance of the electrode. The Hall tests and CV characterizations showed that the TiO_2_/Au/BDD-30 electrode had a better sheet carrier concentration and redox properties than the other electrodes, and thus the TiO_2_/Au/BDD-30 electrode exhibited a better electrochemical oxidation removal efficiency of the reactive brilliant red X-3B. Moreover, the insertion of moderate Au was beneficial to improve the photochemical ability of the electrode by the formation of the Ohmic contact, which could significantly enhance the removal efficiency of the reactive brilliant red X-3B with the introduction of photoirradiation. These findings provided an efficient way to promote the practical application of the BDD electrodes in the field of wastewater treatment.

## Figures and Tables

**Figure 1 materials-16-00868-f001:**
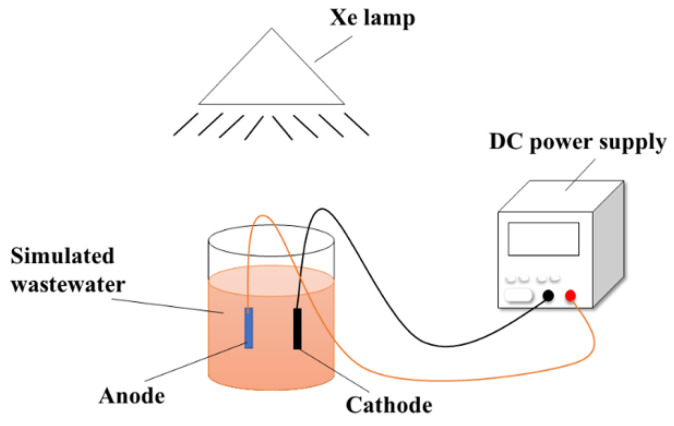
Schematic illustration of the simulated wastewater degradation.

**Figure 2 materials-16-00868-f002:**
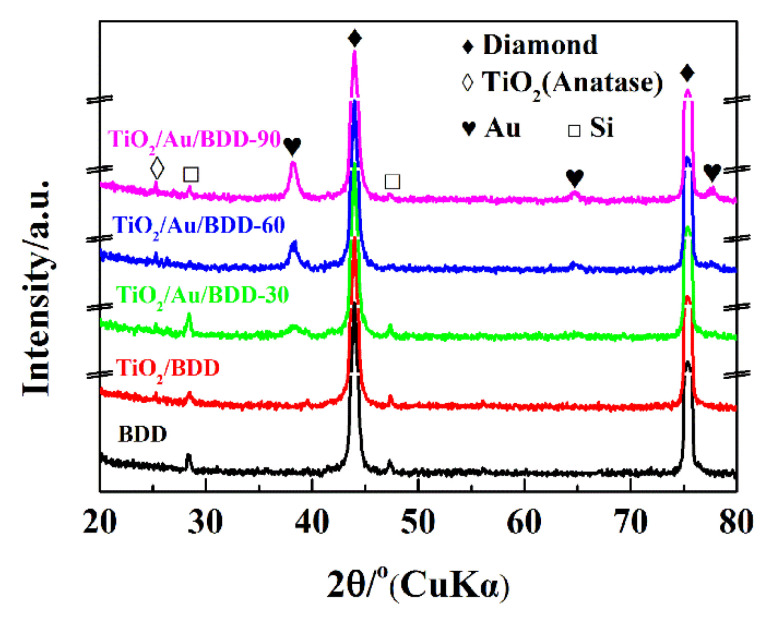
XRD patterns of the BDD, TiO_2_/BDD, TiO_2_/Au/BDD electrodes.

**Figure 3 materials-16-00868-f003:**
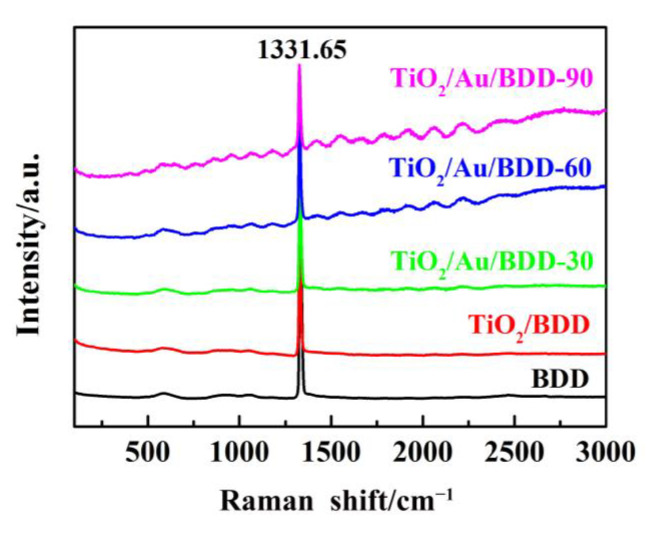
Raman spectra of the BDD, TiO_2_/BDD, TiO_2_/Au/BDD electrodes.

**Figure 4 materials-16-00868-f004:**
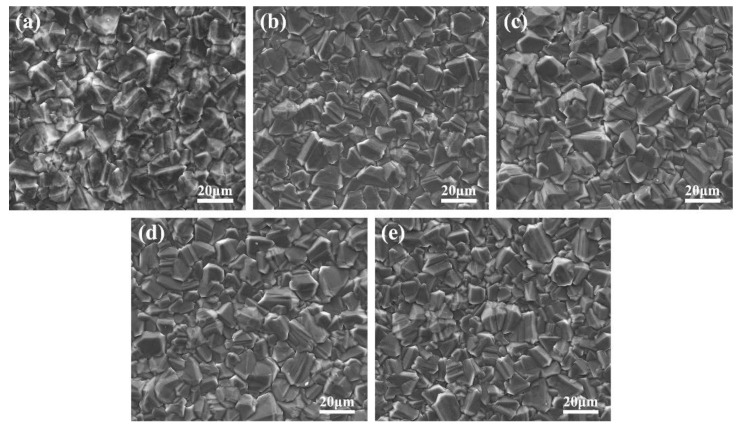
Surface morphology images of the different sample electrodes: (**a**) BDD, (**b**) TiO_2_/BDD, (**c**) TiO_2_/Au/BDD-30, (**d**) TiO_2_/Au/BDD-60, and (**e**) TiO_2_/Au/BDD-90.

**Figure 5 materials-16-00868-f005:**
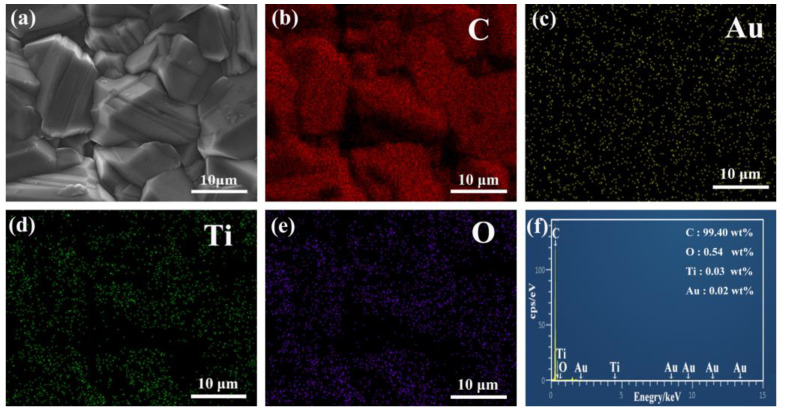
Elemental mapping images of TiO_2_/Au/BDD-30: (**a**) surface morphology of TiO_2_/Au/BDD-30; (**b**–**f**) EDS mapping distribution of TiO_2_/Au/BDD-30.

**Figure 6 materials-16-00868-f006:**
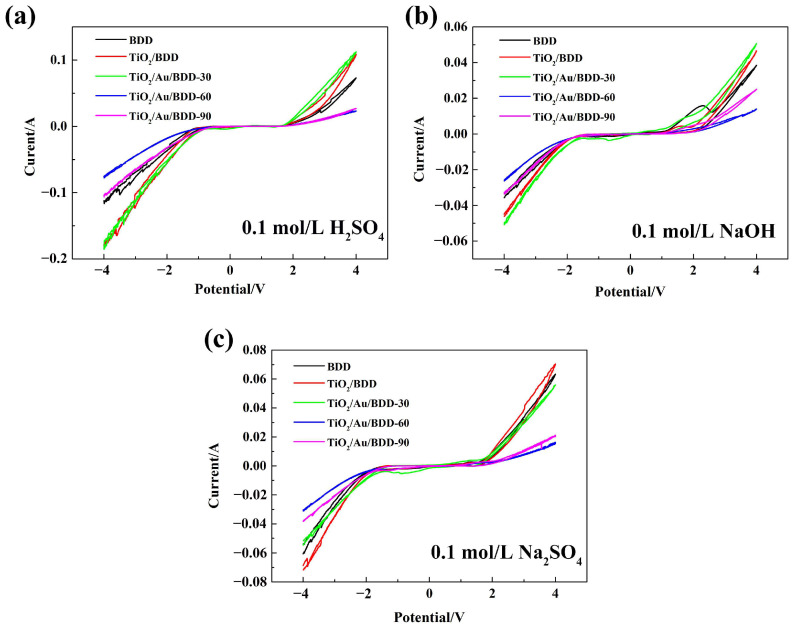
CV curves of the various electrodes in the different electrolytes with the scan rate of 200 mV/s: (**a**) 0.1 mol/L H_2_SO_4_, (**b**) 0.1 mol/L NaOH, and (**c**) 0.1 mol/L Na_2_SO_4_.

**Figure 7 materials-16-00868-f007:**
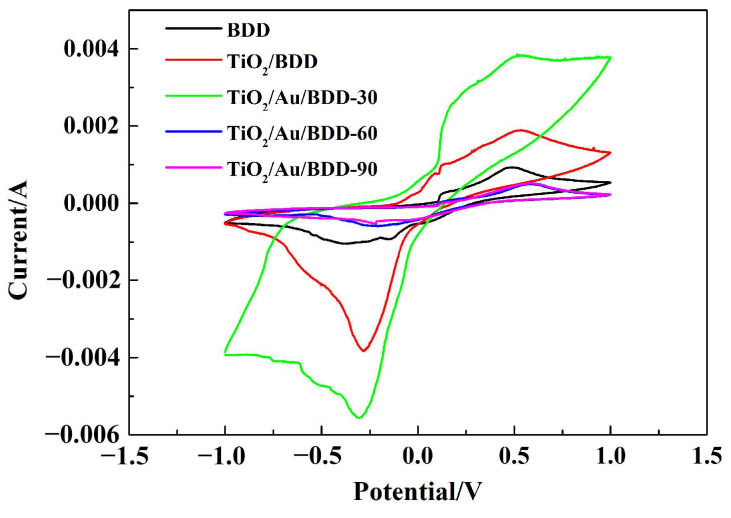
CV curves of the electrodes in 5 mM K_3_Fe(CN)_6_ and 1 M KCl electrolyte.

**Figure 8 materials-16-00868-f008:**
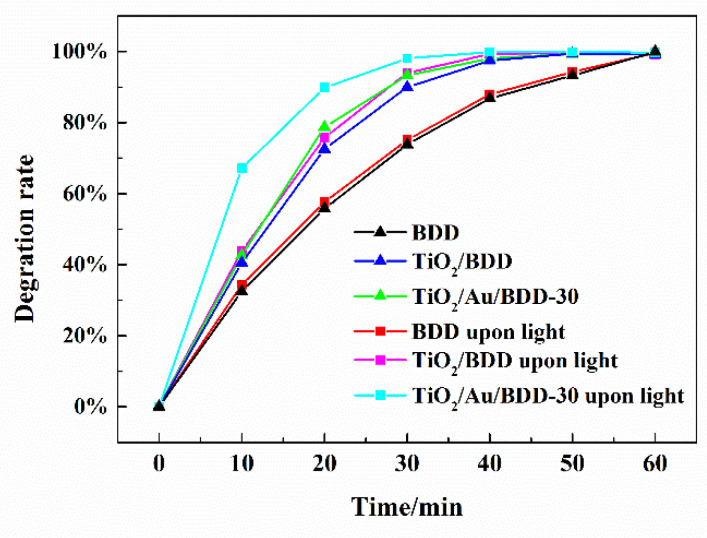
Degradation efficiencies of the simulated wastewater for the various electrodes.

**Figure 9 materials-16-00868-f009:**
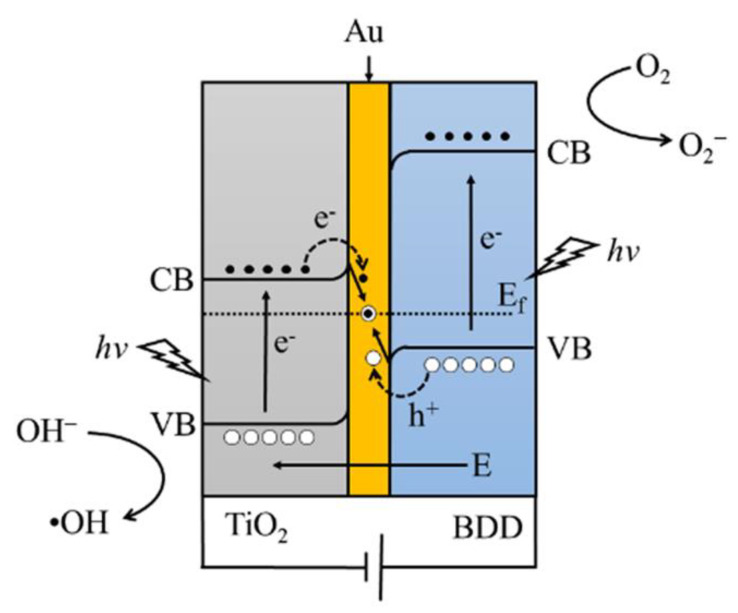
Schematic diagram of the Z-scheme electron transfer in the TiO_2_/Au/BDD electrode.

**Table 1 materials-16-00868-t001:** Hall test results of the different electrodes.

Sample Electrodes	Hall Mobility (cm^2^/V·s)	Sheet Carrier Concentration (cm^−2^)	Sheet Resistivity (Ω·m)
BDD	158	2.26 × 10^17^	0.175
TiO_2_/BDD	178	2.03 × 10^17^	0.173
TiO_2_/Au/BDD-30	194	2.25 × 10^17^	0.143
TiO_2_/Au/BDD-60	239	2.14 × 10^17^	0.122
TiO_2_/Au/BDD-90	486	1.87 × 10^17^	0.069

**Table 2 materials-16-00868-t002:** The comparison of the BDD modified electrodes.

Materials	Oxygen Evolution Potential/V	Potential Window/V	Ref.
Niobium based BDD electrodes	2.57	3.34	[38]
Al_3_BC_3_/BDD electrodes	1.5	1.9	[12]
BDD films electrodes	2.0	2.7	[39]
BDD electrode TiO_2_/BDD electrode TiO_2_/Au/BDD-30 electrode	1.75 1.83 1.56	2.52 2.65 2.16	This work

## Data Availability

Data sharing not applicable.

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
