# Peer review of "Construction of Z-Scheme TiO2/Au/BDD Electrodes for an Enhanced Electrocatalytic Performance"

_materials, 2023, doi:10.3390/ma16020868_

Round 1

Reviewer 1 Report

Comments and Recomendations:

 1. Formula (1) is numbered in the article, but is not mentioned anywhere in the article, i.e. there is no reference to this number. Therefore, there is no need for such a number, it must be removed.

2. The work contains a lot of illustrative material. The drawings are clear, have explanations. But there is no wording of why these drawings are given in the article, how and in what they help in achieving the main goal of the study. And this goal itself is not clearly set. It is necessary to briefly and clearly explain what is the scientific purpose of the work, what exactly is the benefit of this research for other fields of science, is it possible to apply what the authors consider their important scientific result in research in other areas?

3. The fact that gold has disinfecting properties has been known since ancient times. The authors propose a new approach to the use of his abilities. The question is, do the authors have grounds (other than general statements that someone will need it) to believe that their results will be seriously in demand in practice? The opinion of the authors here is especially interesting, because they have access to all the nuances of the question.

This paper is well enough written to understand main results. The manuscript seems to be suitable for publication. I assume therefore that such a work corresponds to the content of the Journal "Materials" and can be published there after minor corrections mentioned above.

Reviewer 2 Report

The authors report the development of composites with Z-scheme structure by depositing gold and titanium dioxide on the surface of boron-doped diamond (BDD) film in order to enhance the use of BDD for wastewater treatment. The reported issues could have relevance in electrocatalysis and in wastewater treatment. However, I believe that major revisions are needed before this manuscript can be considered for publication in this Journal.

- Firstly, the novelty and the advances f the work with respect to that reported by other authors in similar studies must be discussed (see as example Ref. [26] of the manuscript and H. Yu et al., Fabrication of a TiO2−BDD Heterojunction and its Application As a Photocatalyst for the Simultaneous Oxidation of an Azo Dye and Reduction of Cr(VI), Env. Sci.  Technol., 2008, 42, 3791-3796).

- The elemental mapping images in Figure 5 should report the scale bars. Moreover, atomic percentages should be also reported.

- A table reporting the results of electrochemical studies, compared with other literature works should be also inserted.

Round 2

Reviewer 2 Report

I believe that this revised version of the manuscript can be accepted for publication in this Journal.